# OpenReview forum: "SALMON: Self-Alignment with Instructable Reward Models"
_ICLR.cc/2024/Conference — ICLR 2024 poster_

### Official Review · Reviewer_AyPF · 2023-10-29

**Soundness:** 3 good
**Presentation:** 3 good
**Contribution:** 3 good
**Rating:** 6
**Confidence:** 3

**Summary:**

The paper proposes a principle-following reward model trained on a set of human-specified principles, which is then used to learn human-aligned behaviors via the corresponding RL policy.

**Strengths:**

The paper is fairly easy to understand.  The approach seems to be a novel way to align RL policies using LLMs.

**Weaknesses:**

I believe the underlying assumption is that the specified principles (and examplars) are capable of completely specifying aligned behaviors. In reality, this may not be the case, and additionally, specifying an exhaustive list of principles may not be practical. Further results could have also been included (described later).

**Questions:**

1.	It is not entirely clear how the principles and exemplars affect the policy learning. I would have liked to see more experiments where the alignment of policies are evaluated after training with say, just the examples, just the 31 principles, different fractions of the 31 principles, with certain types of principles excluded etc.,

2.	What happens in scenarios where two or more principles are in conflict with each other?

3.	Perhaps related to the previous point, would it be possible to impose a hierarchy of principles during training? I imagine such hierarchies may be important in many practical circumstances.

4.	Is there any way to guarantee that the set of specified principles would indeed lead to an aligned policy? In other words, is the set of principles general enough to be applicable to any scenario?

5.	In pg 7 - It is not clear what a power RLHF trained model is

6.	Pg8 – In 4.1.2 ‘police’ should be ‘policy’

---

> ### Author Response · Authors · 2023-11-22
> **Reply to Reviewer AyPF**
>
> Thank you for your insightful review. We appreciate your recognition of the paper's clarity and the novelty of our approach in aligning RL policies using LLMs. We address your questions below.
>
> > Concern 1: In reality, this may not be the case, and additionally, specifying an exhaustive list of principles may not be practical.
>
> We thank the reviewer for the insightful comment. We have added a new paragraph to discuss the limitation of the Principle Design Challenges, as copied below:
>
> Principle Design Challenges: Crafting robust and encompassing principles for SALMON is intricate, mainly due to the unpredictability of the myriad scenarios a model might encounter
> during the RL stage. Balancing potentially conflicting principles introduces complexities that can yield unexpected results. We advocate for the participation of a diverse group, including
> ethicists and other stakeholders, to refine these guiding principles. It is crucial to recognize that
> distinct contexts and applications will necessitate unique strategies. We present our approach not as a universal solution but as a starting platform, aiming to foster expansive community discourse. In practice, many companies and organizations already have an exhaustive list of principles for their employees, which is usually named as Business Conduct Guidelines (BCG).
>
> On the other hand, we would like to point out that a recent work (published after the ICLR deadline) [1] shows that some general “good-for-humanity” principles can provide high-level guidance to choose outputs that are best for humanity, and achieve similar performance to specific principles. We leave the exploration of SALMON with less amount of general principles as future direction.
>
> > Question 1: It is not entirely clear how the principles and exemplars affect the policy learning.
>
> Please see “Direct Evaluation of the Principle-Following Reward Model” under the General Response for how principles affect the preference of the reward model.
>
> As an approximation of PPO, we also provide the best-of-n samples of how principles affect the results of ranked policy outputs in Section. D of the appendix.
>
>
> > Question 2: What happens in scenarios where two or more principles are in conflict with each other?
>
> As the discriminative preference labels are decided by the principle with the most pronounced difference (Appendix D). The scenarios of conflict principles are determined by the most significant differences between two responses.
>
> > Question 3: would it be possible to impose a hierarchy of principles during training?
>
> We thank the reviewer for the insightful suggestion. We envision that a hierarchy of principles (e.g., safety over helpfulness) is possible and would explore this direction as our future work.
>
> > Question 4: is the set of principles general enough to be applicable to any scenario?
>
> Please refer to our response to Concern 1.
>
> [1]: Kundu, Sandipan, et al. "Specific versus General Principles for Constitutional AI." arXiv preprint arXiv:2310.13798 (2023).

---

> > ### Comment · Reviewer_AyPF · 2023-11-22
> >
> > Thanks for your detailed responses. They, along with your responses to the other reviewers' concerns have helped me better understand the approach.

---

### Official Review · Reviewer_DDqX · 2023-11-01

**Soundness:** 3 good
**Presentation:** 3 good
**Contribution:** 2 fair
**Rating:** 6
**Confidence:** 3

**Summary:**

This paper provides SALMON, a novel method for Large Language Model (LLM) alignment. The main idea of SALMON is self-alignment with principle-following reward models. The current prevailing method for LLM alignment is Reinforcement Learning with Human Preferences (RLHF), and it mainly consists of three phases: (1) supervised fine-tuning (SFT) on human demonstration data, (2) reward mode (RM) training on human preference data, and (3) RL fine-tuning with human-guided RM. Unlike RLHF, SALMON consists of (1) few-shot in-context learning (ICL), (2) RM training on AI preference data, and (3) RL fine-tuning with AI-guided RM. Since SALMON is based on RLAIF, it can be more efficient and scalable than RLHF. More specifically, SALMON based on Llama-2-70B only uses 6 demonstration annotations and zero preference annotations to achieve 7.4 MT-Bench score. In contrast, Llama-2-Chat based on SFT and RLHF uses about 27K demonstration annotations and about 1.4M preference annotations to achieve 6.9 MT-Bench.

**Strengths:**

- S1. First of all, this paper well-written and well-organized.

- S2. It is very interesting that SALMON (one of RLAIF methods) can significantly reduce human annotation costs than a prevalent RLHF method.

- S3. Unlike other RLAIF methods, SALMON can control preference scores by using a principle-following reward model (i.e., changing a principle to follow).

**Weaknesses:**

- W1. One of main contributions of this paper is a principle-following reward model that can control reward scores according to principles. In addition to the overall alignment scores, can the authors measure a quantitative result of the principle-following reward model?

- W2. Even though Llama-2-70B with SALMON can provide better alignment score (7.4 MT-Bench score) than Llama-2-70B with RLHF (PPO) (6.9), there is still large gap to GPT-4 (9.0) and ChatGPT (7.9).

- W3. This paper compares SALMON with PPO-based RLHF. However, enhanced RLHF methods such as DPO (Direct Policy Optimization) and P3O (Pair-wise Policy Optimization) has been proposed and shown that they can achieve better reward score than PPO-based RLHF. It would be better to compare SALMON with recent RLHF methods.

- W4. It would be interesting to provide comparison in perplexity score to see if SALMON is better to maintain the token distribution of the reference LLM than PPO-based RLHF methods.

**Questions:**

- Q1. Regarding W1 above, what is the main advancement of SALMON compared to Constitutional AI?

- Q2. Regarding W2 above, if better base LLMs than Llama-2-70B are used, can SALMON further reduce the gap to GPT-4 and ChatGPT? Or, is SALMON specialized on Llama-2 family LLMs?

- Q3. Regarding W3 above, if some enhanced RLHF methods such as DPO and P3O are used instead of a PPO-based method, is SALMON sill provide better performance than those methods? If not, can SALMON increase its alignment performance by additionally using human demonstration data or AI (or human) preference data?

---

> ### Author Response · Authors · 2023-11-22
> **Reply to Reviewer DDqX**
>
> Thank you for your insightful review and the positive feedback on our paper. We are glad that you found our paper well-written and organized. It's particularly encouraging to hear your appreciation for the practical intuitiveness of SALMON and its efficiency in reducing human annotation costs compared to the prevalent RLHF method. We address your questions below.
>
> > Concern 1: can the authors measure a quantitative result of the principle-following reward model?
>
> Please see “Direct Evaluation of the Principle-Following Reward Model” under the General Response.
>
> > Concern 2: Even though Llama-2-70B with SALMON can provide better alignment score (7.4 MT-Bench score) than Llama-2-70B with RLHF (PPO) (6.9), there is still large gap to GPT-4 (9.0) and ChatGPT (7.9).
>
> We would like to point out that it is unfair to directly compare SALMON with state-of-the-art proprietary models. SALMON uses an open-source LLaMA-2-70B base language model, while GPT-4 uses a much larger and stronger base language model, as evidenced in the LLM capability evaluations (e.g., MMLU, BIG-BENCH). Therefore, we use LLaMA-2-Chat-70b (6.9) as the primary baseline for comparison, and show that SALMON (RLAIF) can outperform the prevailing RLHF method on general alignment.
>
> > Concern 3: However, enhanced RLHF methods such as DPO (Direct Policy Optimization) and P3O (Pair-wise Policy Optimization) have been proposed and shown that they can achieve better reward scores than PPO-based RLHF. It would be better to compare SALMON with recent RLHF methods.
>
> P3O [1] is released to the public in Oct. 2023, before the ICLR deadline. It is impossible for us to compare SALMON against it.
>
> For DPO [2], please see “Compare SALMON with recent RLHF alternatives: DPO, ResT, and SliC-HF” under the General Response.
>
> > Concern 4: if SALMON is better to maintain the token distribution of the reference LLM
>
> We perform the comparison of the aggregated KL-divergence of SALMON with and without RL-time inference intervention, but do not observe significant difference.
>
> |      | Seq. KL-divergence (w.r.t SFT) |
> | ----------- | ----------- |
> | Dromedary-2-70b (PPO w/ RL-time inference intervention)  | 13.37 |
> | Dromedary-2-70b (PPO w/o RL-time inference intervention) | 14.56 |
>
> We would like to point out that the reward-hacking distribution shift in the RLHF training usually does not come with a significant increase in KL-divergence, especially when these RLHF methods have a KL-penalty loss term in the reward (Eq. 2).
>
> > Question 1: what is the main advancement of SALMON compared to Constitutional AI?
>
> Please see our General Response.
>
> > Question 2: if better base LLMs than Llama-2-70B are used, can SALMON further reduce the gap to GPT-4 and ChatGPT?
>
> As a general and scalable alignment paradigm, SALMON is not specialized on LLaMA-2 family LLMs. We believe better base LLMs + SALMON could further reduce the gap to GPT-4 and ChatGPT, as evidenced by the recent works [6,7] showing that RLAIF has a very steep scaling curve (e.g., Figure 4 in [1] and Figure 6 in [2]).
>
> > Question 3: If some enhanced RLHF methods such as DPO and P3O are used instead of a PPO-based method, is SALMON sill provide better performance than those methods?
>
> Since PPO (RLHF) is the only method that has proven effective in large-scale real-world settings [3,4,5], and our method outperforms LLaMA-2-Chat-70b (best PPO-trained non-distilled model) under fair comparison, we believe SALMON would also achieve strong performance with other non-RL (offline RL) PPO alternatives. For example, we found sampling responses from the SFT model, using a principle-following reward model to label the preference, and performing DPO-based optimization can achieve strong performance.
>
> Please see “Compare SALMON with recent RLHF alternatives: DPO, ResT, and SliC-HF” under the General Response for more information.
>
> ---
>
> [1] Wu, Tianhao, et al. "Pairwise proximal policy optimization: Harnessing relative feedback for llm alignment." arXiv preprint arXiv:2310.00212 (2023).
>
> [2] Rafailov, Rafael, et al. "Direct preference optimization: Your language model is secretly a reward model." arXiv preprint arXiv:2305.18290 (2023).
>
> [3] OpenAI. “Introducing ChatGPT” (2022).
>
> [4] OpenAI. “GPT-4 Technical Report” (2023).
>
> [5] Anthropic. “Introducing Claude” (2023).
>
> [6] Bai, Yuntao, et al. "Constitutional ai: Harmlessness from ai feedback." arXiv preprint arXiv:2212.08073 (2022).
>
> [7] Kundu, Sandipan, et al. "Specific versus General Principles for Constitutional AI." arXiv preprint arXiv:2310.13798 (2023).

---

### Official Review · Reviewer_pWc3 · 2023-11-02

**Soundness:** 2 fair
**Presentation:** 3 good
**Contribution:** 2 fair
**Rating:** 6
**Confidence:** 4

**Summary:**

This work proposes SALMON, a method for training reward models that generate scores based on certain guiding principles. First, an instruction-tuned SFT model is used to generate preferences conditioned on a principle. This dataset is then used to train a principle conditioned reward model, where the reward model is trained with many subsets of principles, enabling it to generalize to new principles as well. This instruction tuned reward model is then used in a RLHF loop to fine-tune the SFT model. The resulting model Dromedary-2-70b, tuned from llama-2-70b, shows strong performance on several benchmarks, such as MTBench.

**Strengths:**

- The paper is generally well-written, though addressing some questions related to preference collection should improve the clarity further.
- A relevant and timely problem to address. Preference data needs to be extensively collected to keep reward models in-distribution with the current RL policy.
- The performance of the model is impressive, and the recipe for AI feedback seems quite interesting.

**Weaknesses:**

- Some lack of novelty compared to Constitutional AI; The paper emphasizes constitutional AI focuses more on safety, but the technique itself is very much amenable for building a more “helpful” constitution too. But, the system laid down is distinct enough to warrant interest from the community.

- The paper claims that using principles to avoid reward hacking. Perhaps, the work “reward hacking” is a bit overloaded, but I don’t see any reason that SALMON rewards cannot be hacked to give degenerative responses or undesirable responses, if trained long enough using RL.

- What I do not completely understand is why train a separate reward model at all? The SFT-Model acting as a judge can already be used as a reward model. The scores for SFT-model can be aggregated for many principles as well (it might require multiple passes, but they can be batched potentially).

Overall, it seems that the final reward model ends up using several “hacks” which somewhat go against the main message of the paper:
- The reward model training claims to bypass human preference collection, but ends up pre-training on Anthropic HH-RLHF and SHP preference. Importantly, HH-RLHF and SHP contribute ~320k preference pairs to pre-training, while the SALMON reward uses only ~9.8k prompts (unclear how many preference pairs that yields, given that it can be combined with a large number of principles). How well does SALMON do without without the Preference Model Pre-training?
- Prior works deem the tendency of RLHF to produce lengthier responses as a negative. It is somewhat unfortunate that such a length bonus needs to be explicitly included as a symbolic reward. Can you also elaborate how this reward is included?

While I appreciate the performance of Dromedary-2-70b, the paper lacks several experiments and ablations that give more insight into why the method works. Some quantitative experiments that show how well the SFT model labels the preference in accordance to the principle are severely needed, and ablations of training the model without the “hacks”, and only with the “hacks” would show the importance of SALMON technique.

**Questions:**

- What happens if the answers are indistinguishable based on the principle? For example, when asking for a concise answer (but not necessarily correct, ethical or honest) — would it make sense to have an option for “no preference” when collecting preference from the model?
- For every user prompt and output pairs, are preferences collected using every principle? What is the size of the final preference dataset that is generated by the Self-aligned SFT model?
- Moreover, since the preference is computed based on the difference between logprobs, it would make sense to look at log probability of semantically equivalent answers [1]. The highest log probability can be misleading by itself.
- Why is the preference label decided by the principle with the most pronounced difference? Why not use the sum of scores, for example?
- Can you quantitatively evaluate the preferences generated by the SFT-Model, especially conditioned on the principles? How sensitive is the SFT model to the principle? For example, does using the negative principle flip the preference?
- How does the model perform when using just the SALMON reward without the symbolic bonuses, especially without the length bonus?
- A discussion on more recent works on alignment from scratch can be added such DPO, ReST SliC-HF etc

[1] Surface Form Competition: Why the Highest Probability Answer Isn't Always Right. Holtzmann et al.

---

> ### Author Response · Authors · 2023-11-22
> **Reply to Reviewer pWc3 (1/2)**
>
> Thank you for your constructive feedback on our work involving SALMON and the development of Dromedary-2-70b. We are glad that you appreciate the clarity and relevance of our paper, particularly in addressing the timely problem of preference data collection for reward models. Your recognition of our model's impressive performance and the novel approach to AI feedback is encouraging. We address your questions below.
>
> > Concern 1: Some lack of novelty compared to Constitutional AI
>
> Please see “Novelty Compared to Constitutional AI” under the General Response.
>
> > Concern 2: I don’t see any reason that SALMON rewards cannot be hacked to give degenerative responses or undesirable responses, if trained long enough using RL.
>
> In LLM-based AI alignment, reward hacking, or reward over-optimization [1,2,3], means that an RL-optimized model might exploit certain vulnerabilities in the fixed reward model, thereby artificially boosting its score without genuine performance improvement.
>
> We would like to clarify that reward hacking in LLM alignment can happen with perfect natural language, but often comes with noticeable problematic behavioral traits. For example, in this paper, we identify three types of reward-hacking behaviors in our Figure 3 without degenerated responses. Here, unnatural or degenerated responses would not occur due to either too low reward scores or too-high KL penalty.
>
> Perhaps the degenerated responses would occur if the policy models were trained really long. But in practice, the PPO steps of RLHF/RLAIF on large-enough LLMs are typically within 1000 steps [4], and the degenerated responses do not occur in this short RL period.
>
> > Concern 3: The SFT-Model acting as a judge can already be used as a reward model.
>
> The SFT-Model cannot act as a judge for the following reasons:
>
> Note that at inference time, as opposed to training, all the reward models need to predict a scalar for a single output, without requiring access to its paired output. LLMs are typically better at doing pairwise comparison, but worse at directly assigning a score to a single answer [4,5,6]. Therefore, the SFT-Model in RLAIF is usually only used for labeling AI-generated preferences with pairwise comparison.
> Even for pairwise comparison, the performance of the SFT-Model still falls behind the specially trained reward models. Please see “Direct Evaluation of the Principle-Following Reward Model” under the General Response for more information.
> Aggregating many principles with the SFT-Model, as the reviewer pointed out, can be very inefficient at implementation.
>
> > Concern 4: How well does SALMON do without the Preference Model Pre-training?
>
> We found the SFT-Model generated preferences can be comparable to pre-trained preference models, especially on the helpfulness dimension. Please see “Direct Evaluation of the Principle-Following Reward Model” under the General Response for more information.So SALMON is still possible to achieve strong self-alignment performance without PMP.
>
> However, we would also like to point out that Preference Model Pre-training (PMP) is a common practice in RLHF [8,9] to improve the sample efficiency and the asymptotic performance.
>
> > Concern 5: Prior works deem the tendency of RLHF to produce lengthier responses as a negative.
>
> While some recent work [7] found that “When optimizing for helpfulness, RLHF has been consistently observed to drive models to produce longer outputs”, we have not found any work that claims the lengthier responses as a negative. Instead, [7] argues that longer outputs often prove to be more informative and useful. We are open to further discuss this aspect and would appreciate any additional references the reviewer could provide to enrich our understanding.
>
> > Question 1: What happens if the answers are indistinguishable based on the principle?
>
> Our system does have the “no preference” option when the logprobs of judging either Output (a) and Output (b) as the better option are equal.
>
> > Question 2: What is the size of the final preference dataset that is generated by the Self-aligned SFT model?
>
> The preferences are collected for each principle. We collected a total of 98k = 10 (#principles in Table 6) x 9.8k (#prompts OASST1) principle-specified preferences, and used those to construct around 40k multi-principle preferences.
>
> > Question 3: The highest log probability can be misleading by itself.
>
> We calculate the log probabilities of the SFT-Model for preferring Output (a) or Output (b), not the log probabilities of the answer themselves. The AI-generated preference process is illustrated in Figure 2 (Collecting Principle-Driven Synthetic Preferences).

---

> > ### Author Response · Authors · 2023-11-22
> > **Reply to Reviewer pWc3 (2/2)**
> >
> > > Question 4: Why is the preference label decided by the principle with the most pronounced difference?
> >
> > This design is to encourage the principle-following reward model to notice the most significant differences between two responses.
> >
> > > Can you quantitatively evaluate the preferences generated by the SFT-Model
> >
> > Yes, we added new results to address this question. Please see “Direct Evaluation of the Principle-Following Reward Model” under the General Response.
> >
> > > How does the model perform when using just the SALMON reward without the symbolic bonuses, especially without the length bonus?
> >
> > We compare the Dromedary-2-70b models trained using the SALMON reward model with/without symbolic bonuses on common benchmarks. We found that the symbolic bonuses do not affect the performance on capability benchmarks (e.g.., BBH, HumanlEval). In the chatbot benchmark (MT-Bench), SALMON without the symbolic bonuses can still improve the performance of the SFT model, but underperform the model trained with symbolic bonuses.
> >
> > |      | MT-Score (MT) | MT-Score (T-1) | MT-Score (T-2) |
> > | ----------- | ----------- | ----------- | ----------- |
> > | Dromedary-2-70b (before PPO)      | 6.91 | 7.48 | 6.34  |
> > | Dromedary-2-70b (PPO w/o symbolic bonuses) | 7.12 | 7.43 | 6.80  |
> > | Dromedary-2-70b (PPO w/  symbolic bonuses) | 7.37 | 7.77 | 6.96  |
> >
> > > A discussion on more recent works on alignment from scratch can be added such DPO, ReST SliC-HF etc
> >
> > Please see “Compare SALMON with recent RLHF alternatives: DPO, ResT, and SliC-HF” under the General Response.
> >
> > [1] Skalse, Joar, et al. "Defining and characterizing reward gaming." Advances in Neural Information Processing Systems 35 (2022): 9460-9471.
> >
> > [2] Pan, Alexander, Kush Bhatia, and Jacob Steinhardt. "The effects of reward misspecification: Mapping and mitigating misaligned models." arXiv preprint arXiv:2201.03544 (2022).
> >
> > [3] Gao, Leo, John Schulman, and Jacob Hilton. "Scaling laws for reward model overoptimization." International Conference on Machine Learning. PMLR, 2023.
> >
> > [4] Kundu, Sandipan, et al. "Specific versus General Principles for Constitutional AI." arXiv preprint arXiv:2310.13798 (2023).
> >
> > [5] Bai, Yuntao, et al. "Constitutional ai: Harmlessness from ai feedback." arXiv preprint arXiv:2212.08073 (2022).
> >
> > [6] Zheng, Lianmin, et al. "Judging LLM-as-a-judge with MT-Bench and Chatbot Arena." arXiv preprint arXiv:2306.05685 (2023).
> >
> > [7] Singhal, Prasann, et al. "A long way to go: Investigating length correlations in rlhf." arXiv preprint arXiv:2310.03716 (2023).
> >
> > [8] Askell, Amanda, et al. "A general language assistant as a laboratory for alignment." arXiv preprint arXiv:2112.00861 (2021).
> >
> > [9] Bai, Yuntao, et al. "Training a helpful and harmless assistant with reinforcement learning from human feedback." arXiv preprint arXiv:2204.05862 (2022).

---

### Official Review · Reviewer_7kRD · 2023-11-06

**Soundness:** 4 excellent
**Presentation:** 3 good
**Contribution:** 4 excellent
**Rating:** 8
**Confidence:** 3

**Summary:**

This paper introduces a new self-alignment technique called SALMON, where we can leverage AI feedback with minimal human supervision to align language models with human preference. With those synthetic preference data generated with SALMON, the authors train a new model named Dromedary-2, which achieves state-of-the-art performance on various benchmark datasets.

**Strengths:**

- The methods that the authors propose only need to change the way of generating synthetic data, without much of modification of following RLHF procedure, which makes the technique more general and easy to adapt to other tasks
- The methods can be quite helpful when we need more domain-specific preference data (e.g., code, agents) when there is no such public available data.
- The authors demonstrate the advantage of the new method by finetuning with QLORA on 70B models, demonstrating its ability to improve model performance.

**Weaknesses:**

- It would be good to show that the methods can also be leveraged to improve the performance of smaller models such as 7B or 33B, making the method easier for other topics or tasks.

- I believe this method could potentially be adapted to some other tasks such as code generation. But I am not sure if it is possible, it would be good if the authors could comment on this.

**Questions:**

Please see my questions above.

---

> ### Author Response · Authors · 2023-11-22
> **Reply to Reviewer 7kRD**
>
> Thank you for your insightful review of our paper. We're glad to hear that you appreciate the simplicity and general applicability of our SALMON technique, as well as its potential in domain-specific applications. Your recognition of how our method can generate synthetic preference data with minimal human supervision and its integration with Dromedary-2 to achieve top-notch performance across benchmarks is encouraging. We address your questions below.
>
> > Concern 1: It would be good to show that the methods can also be leveraged to improve the performance of smaller models such as 7B or 33B
>
> Recent works [1,2] show that RLAIF has a very steep scaling curve (e.g., Figure 4 in [1] and Figure 6 in [2]), that is, the model’s discriminative performance is not very far from random guess when the model size is around 22B. On the other hand, the SFT initialization of SALMON, i.e., Principle-Driven Self-Alignment [3], also requires a strong enough base language model (LLaMA-65b/70b). Therefore, we have only experimented with the strongest open-source LLM with 70B parameters.
>
> However, we are optimistic and enthusiastic about compressing the alignment behavior from these large self-aligned models into smaller models, for example, using the Dromedary-2-70b model as the oracle generator/evaluator model to generate distillation data for smaller models. [4] That would be an important direction for deploying these models.
>
> > Concern 2: I believe this method could potentially be adapted to some other tasks such as code generation.
>
> We appreciate the reviewer’s insight. We do believe our SALMON (RLAIF) method can be adapted to more difficult tasks where it’s very hard or costly for humans to annotate preferences. We are actively pursuing this direction as follow-up work.
>
> [1] Bai, Yuntao, et al. "Constitutional ai: Harmlessness from ai feedback." arXiv preprint arXiv:2212.08073 (2022).
>
> [2] Kundu, Sandipan, et al. "Specific versus General Principles for Constitutional AI." arXiv preprint arXiv:2310.13798 (2023).
>
> [3] Sun, Zhiqing, et al. "Principle-driven self-alignment of language models from scratch with minimal human supervision." arXiv preprint arXiv:2305.03047 (2023).
>
> [4] Tunstall, Lewis, et al. "Zephyr: Direct Distillation of LM Alignment." arXiv preprint arXiv:2310.16944 (2023).

---

### Author Response · Authors · 2023-11-22
**General Response**

# General Response

We are grateful to all reviewers for their insightful comments. We appreciate that reviewers found our method to be scalable (7kRD, DDqX) / easy to adapt to other tasks (7kRD) / with impressive performance (7kRD, pWc3), and the paper to be well-written (pWc3, DDqX, AyPF) and timely (pWc3).

Several new experiments and analyses, as per your suggestions, have been incorporated, primarily in the appendices. Additionally, we have addressed each of your questions in the individual responses, and would like to emphasize three particular points here:

## Novelty Compared to Constitutional AI (another RLAIF approach)

We agree both SALMON and Constitutional AI (CAI) [1-3] belong to the RLAIF approach. But we would like to highlight several fundamental differences below.

In terms of the methodology, CAI trains a stand-alone reward model that gives high scores to generally good responses, so its preference cannot be updated once the reward model is trained. In contrast, SALMON trains an instruction-following reward model, to generate reward scores based on customized principles as the preference guideline. This enables us to get full control of the reward model’s preference and alleviate the reward model without online preference collection.
In terms of the research scope, CAI has only been shown to be effective in RLHF-trained models (The starting snapshots of CAI are typically models RLHF-trained only for helpfulness.) and for safety alignment. To the best of our knowledge, SALMON is the first method to answer this pivotal research question: Can RLAIF fully replace RLHF to align language models from scratch in enhancing their general alignment and capabilities?

## Direct Evaluation of the Principle-Following Reward Model

As per reviewers’ suggestions, we conducted some quantitative evaluations of the principle-following reward model. Note that at inference time, as opposed to training, all the reward models can predict a scalar for a single output, without requiring access to its paired output. So the SFT model is prompted with a zero-shot question to choose the more helpful or more harmless answer between A and B. We report the results in terms of accuracy in Table 6 (Appendix C), and report them here:

|                           | Anthropic<br>Helpful | Anthropic<br>Harmless | Anthropic<br>Adversarial |
|---------------------------|----------------------|-----------------------|--------------------------|
| SteamSHP-XL               | 66.8                 | 34.2                  | -                        |
| Open Assistant            | 67.7                 | 68.4                  | -                        |
| LLaMA-2 Safety RM         | 55.4                 | 74.7                  | -                        |
| LLaMA-2 Helpfulness RM    | _72.0_             | _71.0_                | -                        |
| Dromedary-2 PMP RM        | 71.8               | 70.3                  | _56.3_                   |
| Dromedary-2 Safety RM     | 59.2                 | **71.2**              | 15.0                     |
| Dromedary-2 Helpful RM    | **72.1**                 | 64.1                  | 17.3                     |
| Dromedary-2 Adversarial RM| 69.8                 | 66.9                  | **89.8**                 |
| Dromedary-2 Helpful SFT   | 68.4                 | 44.2                  | 20.0                     |
| Dromedary-2 Harmless SFT  | 67.2                 | 54.3                  | 19.1                     |

We found that our reward models can achieve competitive performance to LLaMA-2's reward models that are trained on their in-house preference data, and the human-defined principles can effectively guide the preference of our principle-following reward model. Finally, we find that it is possible to prompt the reward model with new principles to avoid reward hacking, as shown in the Anthropic Adversarial results.

## Compare SALMON with recent RLHF alternatives: DPO, ResT, and SliC-HF

DPO, ReST, SliC-HF are non-RL (or offline RL) alternative optimization algorithms to PPO (RLHF). These works are mostly evaluated on simple tasks like text summarization and machine translation tasks, thus not falling into the category of general alignment. However, we are happy to consider these works as more efficient alternatives to the PPO (online RL) component in SALMON, for example, using our principle-following reward model in ReST, and include them in our future work section.

[1] Bai, Yuntao, et al. "Constitutional ai: Harmlessness from ai feedback." arXiv preprint arXiv:2212.08073 (2022).

[2] OpenAI, “GPT-4 Technical Report”, arXiv preprint arXiv:2303.08774 (2023).

[3] Kundu, Sandipan, et al. "Specific versus General Principles for Constitutional AI." arXiv preprint arXiv:2310.13798 (2023).

---

### Author Response · Authors · 2023-11-22
**Thank you and we are looking forward to your post-rebuttal feedback!**

Dear AC and all reviewers:

Thanks again for all the insightful comments and advice, which helped us improve the paper's quality and clarity.

The discussion phase is about to end soon and we kindly remind the post-rebuttal responses.

We would love to convince you of the merits of the paper. Please do not hesitate to let us know if there are any additional experiments or clarification that we can offer to make the paper better. We appreciate your comments and advice.

Best,

Author

---

### Meta-Review · Area_Chair_UYyj · 2023-12-04

**Metareview:**

While building primarily on existing approaches, the authors present the possibility of reducing reliance on human feedback by using synthetic data for RLAIF. Reviewers have concerns about generalizability (e.g. to other model sizes) and overall methodological novelty (above existing approaches). Given the current -- relevance and the fact that the method seems to address a major issue in the field -- and that the models are quite performant, I think it's worth presenting at ICLR.

**Justification For Why Not Higher Score:**

Overall novelty is not huge

**Justification For Why Not Lower Score:**

important problem / solution that works.

---

### Decision · Program_Chairs · 2024-01-16

Accept (poster)